# The “Crosstalk” between Microbiota and Metabolomic Profile of Kefalograviera Cheese after the Innovative Feeding Strategy of Dairy Sheep by Omega-3 Fatty Acids

**DOI:** 10.3390/foods11203164

**Published:** 2022-10-11

**Authors:** Athina Tzora, Aikaterini Nelli, Anastasia S. Kritikou, Danai Katsarou, Ilias Giannenas, Ilias Lagkouvardos, Nikolaos S. Thomaidis, Ioannis Skoufos

**Affiliations:** 1Laboratory of Animal Health, Food Hygiene and Quality, Department of Agriculture, University of Ioannina, 47132 Arta, Greece; 2Laboratory of Analytical Chemistry, Department of Chemistry, National and Kapodistrian University of Athens, Panepistimioupolis Zografou, 15771 Athens, Greece; 3Laboratory of Animal Nutrition, Faculty of Veterinary Medicine, Aristotle University of Thessaloniki, 54124 Thessaloniki, Greece

**Keywords:** Kefalograviera cheese quality, feeding systems, metagenomics, metabolomics, sheep diet, pasture

## Abstract

The purpose of this study was to examine the effects of two different feeding systems, a control or a flaxseed and lupin diet (experimental), for a sheep flock, on the microbiota and metabolome of Kefalograviera cheese samples produced by their milk. In particular, the microbiota present in Kefalograviera cheese samples was analyzed using 16S rRNA gene sequencing, while ultra-high performance liquid chromatography coupled to quadrupole time-of-flight mass spectrometry (UHPLC-QTOF-MS) was applied to investigate the chemical profile of the cheeses, considering the different feeding systems applied. The metagenomic profile was found to be altered by the experimental feeding system and significantly correlated to specific cheese metabolites, with *Streptococcaceae* and *Lactobacillaceae* establishing positive and negative correlations with the discriminant metabolites. Overall, more than 120 features were annotated and identified with high confidence level across the samples while most of them belonged to specific chemical classes. Characteristic analytes detected in different concentrations in the experimental cheese samples including arabinose, dulcitol, hypoxanthine, itaconic acid, L-arginine, L-glutamine and succinic acid. Therefore, taken together, our results provide an extensive foodomics approach for Kefalograviera cheese samples from different feeding regimes, investigating the metabolomic and metagenomic biomarkers that could be used to foresee, improve, and control cheese ripening outcomes, demonstrating the quality of the experimental Kefalograviera cheese.

## 1. Introduction

Milk and dairy products are classified as one of the most significant food products in terms of their nutritional value and impact on human health. However, various factors can influence the composition of their macro- and micronutrients and quality, including animal traits, farming, and production practices, as well as relative environmental factors. The most important of these factors are the farming system, animal genotype, agro-climatic conditions, and specific farming methods such as milking and feeding practices [1,2,3].

Raw sheep’s milk is a rich and nutritious fluid characterized by high concentrations of protein, fat, minerals, and vitamins compared with milk of other species [4,5]. Furthermore, it supports a consistent, diverse, and complex microbiome involved in the initiation/facilitation of fermentation and metabolome, both of which make multifaceted contributions to human health and the control of spoilage and disease-causing microorganisms [6,7]. Most of the sheep milk produced is used to manufacture cheese, which represents a significant percentage of world agricultural trade [8].

Cheese contains bioactive compounds that can improve consumers’ health and, therefore, the production and feeding strategies to increase the content of these bioactive compounds through the differentiation of its microbiome and metabolome have a special interest. The literature offers abundant information on the factors influencing the composition of milk with animal feeding practices having a pivotal influence [7,9,10]. High-quality pasture can affect the quantity and quality of milk and milk products. The chemical composition of the milk, especially milk fat quality, can be significantly affected by the processing procedures across the dairy production and the feeding system, both of which, in turn, could affect the nutritional value and technological properties of the milk and cheese [11].

Greece traditionally produces a wide range of cheeses, many of which have a Protected Designation Origin (PDO) label [12]. As the cheese microbiota has a fundamental role in cheese making, a better understanding of the composition of these traditional cheeses, as recently carried out for feta [13], will improve their commercialisation. Kefalograviera, another traditional Greek PDO cheese, produced in specific regions (Western Macedonia, Epirus, Aitoloakarnania, and Evritania), by legislated specific processing steps and milk mixture, has a ripening period of at least 3 months. During the last decades, amplicon high throughput sequencing (HTS), metabolomics and other omics-based technologies have gained prominence in the field of dairy science for the traceability and quality assurance of dairy products [14]. A combined foodomics approach based on both targeted metabolomics and metagenomics can be used to elucidate the impact of feeding systems and their influence on milk quality traits. It is the first time in the literature that the combined methodologies of next generation sequencing (NGS) and metabolomics were used to provide evidence of a positive impact of nutritional strategies on sheep milk and Kefalograviera cheese quality.

Culture-independent techniques have a fundamental impact in food microbial ecology, resulting in the consideration of microbial populations as consortia [15]. A breakthrough in this field was the development of high-throughput sequencing (HTS) technologies which entail higher sensitivity concerning the microbial composition compared with traditional culture-dependent methods beyond being considered quantitative [15]. Genes associated with bacterial taxonomy are sequenced through this approach, leading to the phylobiome, the taxonomic profile of the microbial community and the relative presence of its constituents [16].

Despite these advances, metabolomics is being utilized to investigate relative levels, as well as interactions between the metabolite content in a biological system in response to various factors, such as nutrition or interventions [3,17,18,19,20]. The application of metabolomics is being expanding, investigating important factors and biomarker compounds related to animal health, production, authenticity, and contributors to functional properties [3,21]. The interrelationships between cheese microbiota and their metabolites remain largely unstudied. This type of information could be useful if translated into cheesemaking practice; for example, if ripening conditions or the application of adjuncts were adjusted to favor the growth of specific microorganisms associated with the production of desirable flavor compounds.

In a previous study, an omega-3 enriched diet in dairy ewes led to compositional differences in the microbiota determined by conventional microbiological methods and an increase in the omega-3 polyunsaturated fatty acids in the produced Kefalograviera [22]. However, culture-independent methods can provide a more in-depth understanding of the composition and the diet-induced changes in the Kefalograviera microbiota, while metabolomic analysis can provide information on the microbial metabolites. In this study, an integrative analysis was performed to investigate the interrelationships between cheese microbiota and cheese metabolome of two different groups of Kefalograviera cheese samples based on milk produced by sheep with different diets, the control and the experimental with flaxseed and lupin, rich in omega-3 fatty acids, combining 16S rRNA-based microbiota analysis and targeted metabolomics-based workflow using liquid chromatography–high resolution mass spectrometry (LC-QTOFMS). To achieve the maximum metabolomic coverage, two different chromatographic techniques, reversed phase liquid chromatography (RPLC) and hydrophilic interaction chromatography (HILIC), were applied. Data were used for relative quantification across sample groups.

The aim of this research was to identify the key microbiota players and/or metabolites that are characteristic of the control and experimental Kefalograviera cheeses produced by sheep fed with different diets, as well as to determine the interrelations between microbial and metabolic profile. This combined approach can provide a novel insight into the cheese-making processes and the potential for revealing microbial and metabolic biomarkers that could be used to foresee, improve, and control cheese ripening outcomes, and finally the specific quality of the experimental Kefalograviera cheese.

## 2. Materials and Methods

### 2.1. Animals, Dietary Treatments, and Milk Collection

The experimental strategy involved a total of 40 dairy sheep of Frisarta and Chios crossbreeds. The animals were given primarily a soybean meal-based diet named the control diet. A subgroup of twenty animals was withdrawn from the control diet and soybean meal was partially replaced by equal amounts of flaxseed and lupins (20% of the diet, named the experimental diet). The composition of the control and experimental diets are available in Appendix A. The study was carried out in a standardized field setting in Aitoloakarnania (Greece) and lasted two months.

The sampling strategy consisted of a single collection of milk from (a) the bulk tank of the animals that received the control diet (Period 0), one day before they were separated; and (b) the 48 h bulk tank of the animals that received the experimental diet for two months (Period 1) as depicted in Figure 1. The two Kefalograviera cheese types, control and experimental, were produced in a specialized cheese factory, using the two above-mentioned group milk samples as raw material in different batches. Based on the total milk collected from each group, small wheel-shaped Kefalograviera cheese samples were made, namely 11 control and 13 experimental cheese samples (*n* = 24) of different production batches, and were used for the subsequent analyses.

### 2.2. Kefalograviera Cheese Production and Sampling

Sheep’s milk was collected at the producer’s facility and transported to the cheese producer, where it was kept at a temperature of 1–4 °C. Kefalograviera cheeses were manufactured in a certified PDO cheese establishment following standard procedure specific production steps, the addition of rennet, the addition of lactic acid starter culture (a specific mixture of cultures of *Lactococcus lactis* subsp. *lactis*, *Lactococcus lactis* subsp. *cremoris*, *Streptococcus thermophilus*, *Lactobacillus bulgaricus*, and *Lactobacillus rhamnosus*) and a ripening period for 3 to 6 months. All samples were transported and kept under refrigeration (≤4 °C) before processing, while tested after 6 months of ripening.

### 2.3. DNA Isolation

Ten-gram samples of cheese were collected from within a block of cheese at a depth of approximately 5 mm. The samples were aseptically weighed on one side of sterile filter stomacher bags (BioMérieux (UK) Ltd., Basingstoke, UK). Ninety milliliters of buffered peptone water were added, and the samples were homogenized in a stomacher (Laboratory Blender Stomacher 400; Seward, London, UK) for 2 min at 260 rpm. Ten milliliters of filtered homogenized sample were collected to a 15 mL conical centrifuge tube and high-quality metagenomic DNA was extracted using DNeasy PowerFood Microbial kit (Qiagen, Hilden, Germany) according to the manufacturer’s instruction. DNA concentrations were measured using a fluorescence spectrometer (Qubit, Life Technologies, Carlsbad, CA, USA). The samples were stored at −20 °C.

### 2.4. Sequencing Preparation, Run, and Processing

As sequencing is an expensive methodology, a subset of cheese samples, 9 control and 4 experimental cheese samples (*n* = 13), was used for the microbiota composition analysis. The difference in the number between control and experimental cheese samples can be explained by the intended use of control samples as reference in another study and they were therefore produced in higher quantity. The same samples were also used for the subsequent correlation analysis. The library preparation of the 16S rRNA gene amplicon (V3–V4 hypervariable regions targeted using 341F and 806R primer pair) and its pair-end sequencing on the Miseq platform (Illumina, San Diego, CA, USA) were carried out as described previously [23].

#### Data Processing—Operational Taxonomic Units (OTUs) Analysis

The 16S rRNA gene amplicon data was analyzed and raw reads were merged using the NGS toolkit and further processed using the “Integrated Microbial Next-generation sequencing” (IMNGS, www.imngs.org, accessed on 5 April 2022) pipeline [24] based on UPARSE. Rhea [25] was used to determine α- and β-diversity and bacterial OTUs in an R programming environment (R i386 3.6.0, R Foundation for Statistical Computing, Vienna, Austria). A detailed description of the analysis and the scripts is available online (https://lagkouvardos.github.io/Rhea//, accessed on 7 May 2022). *p*-values were corrected for multiple comparisons and all given results were statistically tested with the Wilcoxon rank-sum test and/or Kruskal−Wallis Rank Sum Test, unless stated otherwise. Significant OTUs were then identified by EzBioCloud’s 16S rRNA gene-based ID. Data were visualized using Illustrator CS6 Version 16.0.0 (Adobe Inc., San José, CA, USA).

### 2.5. Chemicals and Reagents

All solvents for LC-HRMS were of high-purity (UPLC-MS grade). Methanol (MeOH) and acetonitrile (ACN) were purchased from Merck (Darmstadt, Germany), whereas sodium hydroxide monohydrate (≥99.9%), ammonium formate (A.F.) ≥99.0%, ammonium acetate (A.A.) and formic acid (F.A.) 99%, were provided from Fluka (Buchs, Switzerland). Milli-Q purification apparatus (Millipore Direct-Q UV, Bedford, MA, USA) was used for distilled water. Analytical standards utilized in the study were of highest available purity grade. Most of them were purchased from Sigma-Aldrich (Steinheim, Germany). Additional standard solutions used for the in-house database development were prepared according to Mass Spectrometry Metabolite Library of Standards (MSMLS) protocols (more information in SM-2). A working mix solution of 5 mg L^−1^ was prepared for analysis by gradient dilution of the stock solutions in methanol, in order to prepare the matrix-matched calibration curves. Lysine-d4, syringaldehyde and hesperetin were used as Internal Standards (IS) at a level of 10 mg L^−1^.

### 2.6. Metabolomics Sample Preparation Protocol

For the efficient extraction of cheese metabolic content in terms of the coverage of different metabolite’s polarities, a solid–liquid extraction was performed combining different extraction solvents, following a two-step defatting procedure. The procedure was based on an optimization of a sample preparation procedure reported in a previous study by the group [26]. Briefly, the fresh Kefalograviera cheese samples were lyophilized (at −55 °C, 0.05 mbar) using a LyoQuest-55 laboratory freeze dryer (Telstar) and homogenized prior to analysis. For the extraction, 1 g of each freeze-dried sample was weighed in a 15 mL centrifuge tube. Weighted samples were spiked with appropriate concentrations of target compounds and IS used in the study. Following spiking, all the spiked samples, before further analysis, were allowed to stand for 10 min. Addition of 2 mL of H_2_O to the samples was performed, followed by the addition of 2 mL of MeOH and 2 mL of ACN, in order to extract the metabolites and precipitate the proteins. Vortex mixing of each tube for 30 s was carried out after the addition of each solvent. Afterwards, the samples were placed in an overhead shaker for 30 min to improve the extraction efficiency. Ultrasonic extraction for 20 min at 40 °C was performed for all samples followed by centrifugation for 5 min at 4000 rpm. The supernatants were placed into new tubes and then were kept at −23 °C for 12 h (overnight) for the precipitation of lipids and remaining proteins. The samples were again centrifuged in order to remove the precipitate. For the defatting step, the supernatant was transferred to another tube, 3 mL of hexane was added, vortexed for 1 min, and then centrifuged under the same conditions. The hexane layer was removed and a second defatting step was further performed using 3 mL of petroleum ether. Each extract was vortexed again for 1 min, and then centrifuged under the above-mentioned conditions. The final extracts were evaporated to dryness under a nitrogen stream. The temperature must not exceed 40 °C. Reconstitution of remaining residues in 0.5 mL of methanol/water (80:20 *v*/*v*) was carried out for all samples, while filtration was performed through 0.22 mm RC filters. For the preparation of the matrix-matched standard solutions required for the analysis, multi-analyte solutions were added to blank aliquots and vortex mixed for 10 s. For the LC-HRMS acquisition, the reconstituted extracts were put into appropriate vials, where 5 μL were injected into the instrument.

### 2.7. LC-ESI-QTOFMS Instrumentation

An ultra-high performance liquid chromatography (UHPLC) system with an HPG-3400 pump (Dionex Ultimate 3000 RSLC, Thermo Fisher Scientific, Dreieich, Germany) coupled to a quadrupole time-of-flight mass spectrometer (QTOF) (Maxis Impact, Bruker Daltonics, Bremen, Germany) was used for the analysis of the metabolic content of the samples. Samples were analyzed using reversed phase liquid chromatography (RPLC). Hydrophilic interaction liquid chromatography (HILIC) was used as a complementary separation technique, using electrospray ionization interface (ESI), operating in positive (PI) and negative ionization (NI) mode, in both separations.

In RPLC analysis, an Acclaim C18 column (2.1 × 100 mm, 2.2 μm) from Thermo Fisher Scientific (Dreieich, Germany) preceded by a C18 guard column (at 30 °C) was used for the chromatographic separation. In PI, mobile phases consisted of H_2_O/MeOH 90/10 (*v*/*v*) (solvent A) and MeOH (solvent B), both containing 5 mM A.F. and 0.01% F.A. In NI, the same solvents were used, both containing 5 mM A.A. For the elution, a gradient elution program was used, same at both ionization modes. Program started with 1% B (flow rate of 0.2 mL min^−1^) for 1 min, increased to 39% within 2 min and then to 99.9% (flow rate of 0.4 mL min^−1^) for 11 min. For 2 min, 99.9% B was kept for 2 min (flow rate of 0.48 mL min^−1^), while initial conditions were restored within 0.1 min, for the next 3 min; then the flow rate decreased to 0.2 mL min^−1^.

In HILIC analysis, metabolites were separated using an ACQUITY C18 BEH Amide column (2.1 × 100 mm, 1.7 μm) purchased from Waters (Dublin, Ireland) preceded by a C18 guard column, thermostatted at 40 °C. For the PI, mobile phases consisted of H_2_O (solvent A) and ACN/H_2_O 95/5 (*v*/*v*) (solvent B) both amended with 1 mM A.F. and 0.01% F.A. For the NI, in both mobile phases, 10 mM A.A. were added. The adopted gradient elution program, for both ionization modes, started with 100% B for 2 min, decreased to 5% in 10 min and was kept stable for 5 min. Restorage of initial conditions were performed within 0.1 min and then re-equilibration of the column was performed for the next 8 min. Flow rate was 0.2 mL min^−1^.

Regarding mass spectrometry, the QTOF-MS system provided an ESI source, operating in PI and NI, according to the following instrumental conditions: capillary voltage 2500 V (PI) and 3500 V (NI); end plate offset 500 V; nebulizer pressure 2 bar; drying gas 8 L min^−1^; and gas temperature 200 °C. The QTOF-MS system was operated in data-dependent acquisition (DDA) mode (AutoMS/MS), as well as in data independent acquisition (DIA) mode (broadband collision-induced dissociation, bbCID. Mass range for the metabolites detection was set at *m*/*z* 50–1000 (scan rate: 2 Hz). For the verification of the analytical performance, external instrumental calibration was performed in every run according to the manufacturer’s guidelines.

### 2.8. LC-HRMS—Metabolomics Data Processing Workflow

#### 2.8.1. Target Screening Workflow

For the evaluation of the data acquired from LC-HRMS analysis, a target screening metabolomics approach was applied. Specifically, Data Analysis 4.3 and TASQ 2.1 software (Bruker Daltonics, Bremen, Germany) were initially utilized for the data evaluation. Four different in-house metabolite databases (RPLC (+): 208 compounds, RPLC (−): 164 compounds, HILIC (+): 179 compounds, HILIC (−): 144 compounds) were exploited comprised of different classes of metabolites, as amino acids, sugars, fatty acids, etc., and their derivatives. Each database included important characteristics for the compounds, such as molecular formulas, retention time (min), pseudomolecular ions, as well as MS/MS fragments (qualifier ions) (Appendix A, Appendix A). The identification was performed according to specific identification criteria, as reported in a previous study by the group [27] (SM-1). Most of the analytes were submitted to quantification through the preparation of matrix-matched standard calibration curves (SM-2.1).

#### 2.8.2. Statistical Analysis

Multivariate statistical analysis, including principal component analysis (PCA), was applied through an in-house developed workflow in R environment (R Studio, Version 1.1.463, Boston, MA, USA) and autoscaling was applied. The unsupervised PCA was used in the study as an initial descriptive approach to the obtained LC-HRMS data, in order to investigate any existing clustering in cheese samples based on their metabolite content [28].

### 2.9. Correlation Statistical Analysis

Combinations of variables that are connected with linear relationships were detected by calculating their Pearson’s coefficient of correlation. The centered log-ratio transformation was used to remove the compositional constraints from the taxonomic variables. In addition, taxonomic zeros (relative abundance of taxonomic variables with the value zero) have been treated as missing data and excluded from the calculation of correlations. Following this transformation of taxonomic variables, the table was centered and scaled, to adjust for differences in the offset and fold changes respectively, and the Pearson correlation for all pairs was calculated. The significance before and after FDR correction was reported together with the number of observations that supported the correlation [29,30,31].

## 3. Results and Discussion

### 3.1. Sequencing Coverage

#### 3.1.1. α-Diversity

We extracted a total of 276,500 filtered sequence reads from the 13 samples and obtained 268,748 high-quality sequences. The average sequence number for each sample was 17,381 (ranging from 3373 to 21,911). The bulky of microbial diversity has been captured as shown by the rarefaction curves. α-diversity calculated through both the Simpson and Shannon indices and their effective numbers (Figure 2).

#### 3.1.2. β-Diversity

A measurement of the similarity between different microbial profiles described by the OTUs between the two groups was calculated through a generalized UniFrac. In Figure 3, a visualization of the multidimensional distance matrix in a space of two dimensions was performed by multi-dimensional scaling, revealing a significantly different microbial community of the control vs. the experimental group cheese samples at both the taxonomic genus and family level.

### 3.2. Bacterial Diversity Estimation

We can achieve a more in-depth characterization of the microbiota of the Kefalograviera cheeses using HTS techniques in order to optimize their quality, safety, and commercial values. Since scarce data is available using HTS approaches in Kefalograviera cheese production, this methodology was used in this study as a systematic approach to characterize the microbial composition and richness in response to the different animal feeding systems.

The predominant members of the Firmicutes phylum in Kefalograviera cheese-associated microbiota accounting for approximately 99.8% of the entire abundance, while Proteobacteria and Actinobacteriota were solely present in only one experimental cheese. The overall bacterial composition showed a diversity with 9 families (Figure 4A) constituting the vast majority of the community: *Streptococcaceae* (relative abundance: 65.99%), *Lactobacillaceae* (relative abundance: 32.99%), *Leuconostocaceae* (relative abundance: 0.72%) and others (*Enterobacteriaceae*, *Enterococcaceae*, *Intrasporangiaceae*, *Pseudomonadaceae*, *Staphylococcaceae*, *Xanthomonadaceae*).

At the genus level, *Streptococcus*, *Lactobacillus*, *Lacticaseibacillus*, *Lactococcus*, *Pediococcus* and *Leuconostoc* constituted more than 90% of the total bacterial population and, therefore, the subsequent analysis focused on determining the dominant species of these genera (Figure 4B). Among *Streptococcus*, *Streptococcus thermophilus* represented most of the total population, accounting for >98% of total relative abundance of this genus in both experimental and control cheese samples. Among *Lactobacillus*, most species were identified as *Lactobacillus delbrueckii* subsp. *bulgaricus* (7.68% of total relative abundance on average), *Lacticaseibacillus zeae*/*casei*/*paracasei*/*chiayiensis* (about 18.99% of total relative abundance on average), *Lactobacillus helveticus*/*crispatus*/*gallinarum* (about 4.67% of total relative abundance on average) and *Levilactobacillus brevis* (about 0.26% of total relative abundance on average). *Lactococcus* population was dominated by *Lactococcus lactis*/*cremoris* (about 6.36% of total relative abundance on average) and *Pediococcus* was composed of about 1.23% of total relative abundance on average. To our knowledge, this is the first description of the Kefalograviera microbiota using HTS technologies. The results presented in this study are mostly in disagreement with a recent publication on Greek cheeses including Kefalograviera [32]; however, that study used different molecular methodologies and focused on the lactic acid population with variation in starter culture composition, cheese-making procedures, raw milk used and more probably accounting as additional factors for the differences between the two studies.

It is worth noting that except for the high abundant OTUs at species level presented for both control and experimental cheeses, an increase in *Pediococcus pentosaceus* and *Leuconostoc falkenbergense* was additionally observed in the experimental cheese samples; however, it was not statistically significant (*p*-value > 0.05). Several reports have indicated that pediococci found in non-starter population of milk and play a key role in the flavor development of cheese during the ripening process by the enzymatic degradation of amino acids which are important precursors for flavor compounds [33]. Concerning *Leuconostoc* spp., these bacteria are involved in mannitol production by a dehydrogenase that catalyzes the reduction of fructose to mannitol, while the ability to produce bacteriocins is an additional interesting property attributed to some strains of this genus [34].

The bioinformatics analysis showed an increase in the effective richness of the experimental cheese samples indicating differences in the biodiversity of the two types of Kefalograviera [35]. With regard to species-level composition, the OTUs that showed high and significant differences in relative abundance (%) between the two groups of cheeses (control and experimental cheeses) were OTU_4: *Lactobacillus delbrueckii* subsp. *bulgaricus* (*p*-value = 0.0198) with 100% similarity identified by EzBioCloud’s 16 S rRNA gene-based ID and OTU_21: *Lactobacillus helveticus*/*crispatus*/*gallinarum* (*p*-value = 0.0202) with 99.1% similarity identified by EzBioCloud’s 16 S rRNA gene-based ID that decreased and increased in the experimental cheese samples, respectively (Figure 5). In our previous study, the omega-3-enriched diet of dairy ewes influenced the lactobacilli population of the produced Kefalograviera [22]; however, different *Lactobacillus* species were affected, namely, *Lb. rhamnosus*, *Lb. plantarum* and *Lb. paracasei*, which can probably be attributed to the different methodology (culture-dependent technique) used for the sample analysis.

Taking all the above together, the analysis of the microbial community in experimental and control cheese samples revealed that the different feeding system influenced the bacterial composition in the produced cheese, highlighting the important role of animal diet in the cheese-making process.

### 3.3. Target Metabolomic Screening

Through the UHPLC-ESI-QTOFMS analysis, a targeted metabolomics-based strategy was applied to thoroughly screen and profile low-molecular-weight metabolite content in various cheese samples. Four in-house target databases were used to screen and quantify the content of metabolites in all the studied cheese samples. The list of compounds for RPLC positive contained 208 metabolites, while the list for RPLC negative contained 164 metabolites. In case of HILIC, the list of compounds for positive ionization mode comprised of 179 metabolites, while the list for negative mode contained 144 metabolites. Following the reported data treatment-processing workflow, screening was carried out using TASQ Client 2.1 and the detected compounds were identified according to the suitable processing parameters regarding mass accuracy, retention time, diagnostic ion detection and observed isotopic patterns, as reported above. Internal calibration was performed in all the data retrieved. Data analysis 4.3 Software was used as an additional step of confirmation of identification. Figure 6 represents the workflow used in the study to screen and detect metabolites in cheese extracts. In target screening, the compounds detected in cheese samples consisted of 126 metabolites out of 209 of the initial lists of compounds in RPLC positive, 85 metabolites out of 165 in RPLC negative, 98 metabolites out of 180 in HILIC positive and 43 metabolites out of 145 in HILIC negative (Figure 6A).

Common compounds were observed among the RPLC and HILIC chromatography in positive and negative ionization mode. For quantification purposes, a further evaluation was performed in terms of sensitivity and chromatographic performance to reach high identification levels for most of the compounds and adequate analytical performance. Data Analysis 4.3 Software was used for this analytical process and the evaluation was based on the recorded retention time, peak area of the chromatographic peak and MS fragments of the common compounds. (Figure 6B) represents an example of the compound L-ornithine, which was eluted in both RPLC and HILIC chromatography, in positive ionization mode. For the same sample, it can be observed that, in terms of retention time, L-ornithine was eluted in 9.6 min, while in RPLC in 1.3 min, close to the void volume of the chromatographic column. In addition, in terms of sensitivity, it was observed that in the MS spectrum, the intensity of the precursor ion was higher in HILIC (3 × 10^5^) than in RPLC (2 × 10^5^). Additionally, in RPLC no fragments of the precursor ion were detected, leading to decreased identification level. Thus, for this metabolite, HILIC was proved to be more reliable for quantification purposes and the most efficient chromatography for compounds eluted in both RP/HILIC was chosen for further quantitative results. Finally, a total of >120 detected metabolites identified in high confidence levels are summarized in Appendix A. In particular, various classes of metabolites were annotated using the in-house databases. Most of the metabolites detected belong to specific chemical classes, presented in Figure 6C, while the significant number of compounds identified by this approach is directly linked to the complexity of the food matrix investigated. More than 80 metabolites fully met the identification criteria, showing high mass accuracy (<5 ppm), acceptable isotopic fit values (<100 mSigma) and an acceptable retention time window (0.2–0.4 min). The level of identification of the 85 compounds was Level 1 [36], as they were successfully identified, and their structures were confirmed with reference standards (Appendix A). Thirty-eight metabolites showed lower ion intensities (<6.000) and did not show any fragments in the MS/MS spectra. In this case, the level of identification of these compounds was Level 3, as the MS/MS spectra were not informative enough to proceed in further identification. These metabolites are included in SM-1.

### 3.4. Comparison of Metabolite Profiles between the Different Groups

For the majority of detected and identified metabolites, quantification results were provided by the approach of matrix-matched standards. Specifically, more than 50 compounds were quantified for the two groups (SM-1, Appendix A), investigating significant differences between the groups in concentration levels. With a quick overview, it can be observed that some of the metabolites were in higher concentrations in the experimental samples, except for some cases.

Using the quantification data calculated for the majority of detected and identified metabolites for each group of samples, an unsupervised PCA was constructed to first assess their distribution and determine whether the cheese samples from various animal diets fell into the same cluster groups (Figure 7A). According to this figure, samples from the same class were grouped together, demonstrating effective class discrimination. The three PCs also accounted for more than 40% of the variance. This is a preliminary indication that any PC-based supervised approaches, such as PLS-DA model, are likely to be successfully applied [37]. (Appendix A).

Characteristic analytes that present significant differences between groups are arabinose, dulcitol, hypoxanthine, itaconic acid, L-arginine, L-glutamine and succinic acid. Figure 7B represents the variance of these metabolites between the groups. With a quick overview, it can be observed that all the metabolites were in higher concentration in the experimental samples, except for L-arginine, Figure 7C. However, the strongest differential metabolite was succinic acid, as its average concentration was approximately three times higher in the experimental samples Figure 7D.

The human metabolome database (available at https://hmdb.ca/, accessed on 19 August 2022) was used for the interpretation of the significantly different metabolites between the control and experimental cheese samples. L-arabinose is a bio-active compound and belongs to the class of aldopentoses. L-arabinose is found in all organisms from bacteria to plants to animals. There are two different arabinose utilization pathways in nature: bacterial and fungal. Arabinose has a sweet taste and is one of the most abundant components released by complete hydrolysis of non-starch polysaccharides (NSP) of vegetable origin. L-arabinose is known to selectively inhibit intestinal sucrase activity in a non-competitive manner. Sucrase is the enzyme that breaks down sucrose into glucose and fructose in the small intestine. As a result, L-arabinose suppresses plasma glucose increase due to sucrose ingestion. Indeed, this natural sugar has the taste and natural properties of sucrose, but it is an anti-hyperglycemic factor known to reduce symptoms associated with type 2 diabetes. It inhibits the hydrolysis of sucrose into glucose and fructose. Studies have shown that replacing sucrose with L-arabinose is potentially a good strategy to lower glycemic and insulin responses. Dulcitol, a sugar alcohol takes part at the galactose metabolic pathway while it is a key sugar driving the microbial and metabolic diversity. Additionally, hypoxanthine can improve the anti-microbial properties of the milk against pathogens by increasing endogenous milk xanthine oxidase activity while the itaconic acid is totally produced by microbial fermentation in cheese samples. The higher concentrations of L-arginine can be explained by the fact that it is an amino acid which can be found in soybean in high concentrations while succinic acid is a compound, that can be found not only in soybean, but also in flaxseed and lupins in high concentrations, which were constituents of the experimental animal diet. According to literature, succinic acid, L-glutamine and hypoxanthine have also been reported as discriminant metabolites among different ripening times in cheese samples by Le Boucher, et al. [17]. Succinic acid was also a discriminant metabolite between mare’s milk samples and koumiss samples in the study of Xia, et al. [38].

### 3.5. Correlating Metabolomic and Metagenomic Profiles

Significant metabolomic differences have been observed between the control and experimental cheese samples (Figure 8A) with only the variables being significant in the Kruskal–Wallis test being plotted. Ten metabolites showed higher concentrations (μg/g) in the experimental cheese group. These metabolites were L-glutamine, tryptamine, uracil, N-methyl-L-glutamic acid, itaconic acid, palmitate, succinic acid, 2-oxoadipic acid, L-homoserine and L-proline.

Focusing on specific metabolites, L-glutamine is included in protein-rich foods such as beef, chicken, fish and dairy products, while it is often used as a dietary supplement. Tryptamine is strongly related to specific bacterial families. It can be detected in several different foods, and this could make tryptamine a potential biomarker for the consumption of these foods. L-homoserine is also a microbial metabolite produced during the fermentation process. These interpretations were based on the human metabolome database (available at https://hmdb.ca/, accessed on 19 August 2022). Regarding the other statistically significant metabolites, these have already been described in Section 3.4.

Overall, Figure 8B reported a summary containing those OTUs establishing correlation with the metabolomic compounds. At the OTUs level, OTU1 established strong negative correlation with the microbial diversity (Shannon effective). Concerning OTU3, a negative correlation was remarked with sphinganine and thymine. Additionally, OTU5 showed a strong positive correlation with 4-imidazoleacrylic acid, while a negative correlation with L-phenylalanine was observed. Finally, the OTU21, was the OUT showing the most correlations with the abovementioned metabolites. Specifically, positive correlations with arabinose, azelaic acid, L-glutamine, uracil, N-acetyl-L-proline, N-methyl-L-glutamic acid, itaconic acid, succinic acid and 2-oxoadipic acid were observed. Pearson’s correlation coefficients were considered for significant marker compounds (from metabolomic analysis) and microbial families and genera (from targeted metagenomic analysis) (Figure 8C). To our knowledge, this is the first report of these bacterial OTUs being associated with the above-mentioned metabolites. Further research could potentially provide a better understanding of the importance of the identified correlations in cheese quality and possible impact on human health.

## 4. Conclusions

In this work, a targeted metagenomic and metabolomic approach based on 16s rRNA sequencing and ultra-high-performance liquid chromatography coupled with quadrupole time-of-flight mass spectrometry (UHPLC-QTOFMS), followed by multivariate statistics, was carried out to discriminate Kefalograviera cheeses according to their microbial and chemical profiles. The utilization of metagenomic and metabolomic-technologies allowed us to detect unique microbial and chemical signatures linked to the dietary intervention. To find the important and cheese origin-specific metabolites, target screening was used in accordance with strict identification criteria. More than 120 metabolites were found and identified, including those related to proteolysis products as well as amino acids and their derivatives; organic acids, lipids and their derivatives; fatty acids, vitamins, and nucleotides. Quantification data were also provided for the majority of metabolites, in order to evaluate their differentiation between the control diet compared to the flaxseed and lupin diet. The main differences identified were distinct key microbial players such as *Pediococcus pentosaceus*, *Leuconostoc falkenbergense*, *Levilactobacillus brevis* and arabinose, dulcitol, hypoxanthine, itaconic acid, L-arginine, L-glutamine and succinic acid as metabolomic markers. In this study, we clearly demonstrated that metabolomics and targeted metagenomic analysis combined with statistical methods have great potential for distinguishing between innovative Kefalograviera cheese production systems, differentiated by the animal diet. Metabolomic and metagenomic methods can also be used to identify and analyze different compounds and taxonomic units in fermented dairy products, which could serve as potential biomarkers, indicating a novel traceability system and the authenticity of the final product. Although further studies are needed to strengthen the viability of metabolomics and metagenomics followed by multivariate statistics (including validation of the markers identified), current preliminary results are encouraging in the field of cheese authenticity and traceability.

## Figures and Tables

**Figure 1 foods-11-03164-f001:**
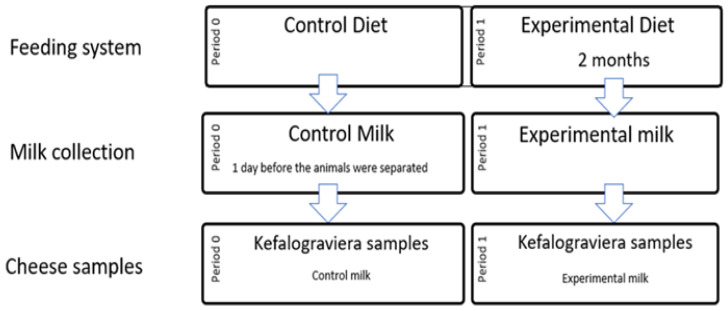
Schematic representation of the sampling strategy used for the microbial characterization of Kefalograviera cheese samples.

**Figure 2 foods-11-03164-f002:**
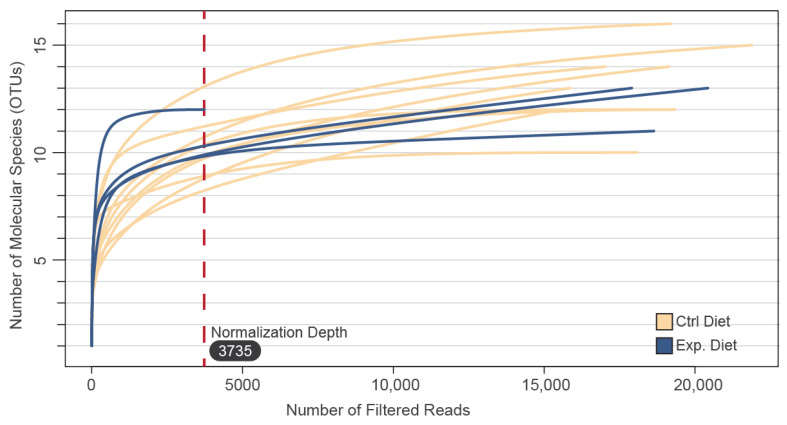
Rarefaction curves for OTUs calculated in Kefalograviera cheeses from 2 groups (Control Diet; Experimental Diet.

**Figure 3 foods-11-03164-f003:**
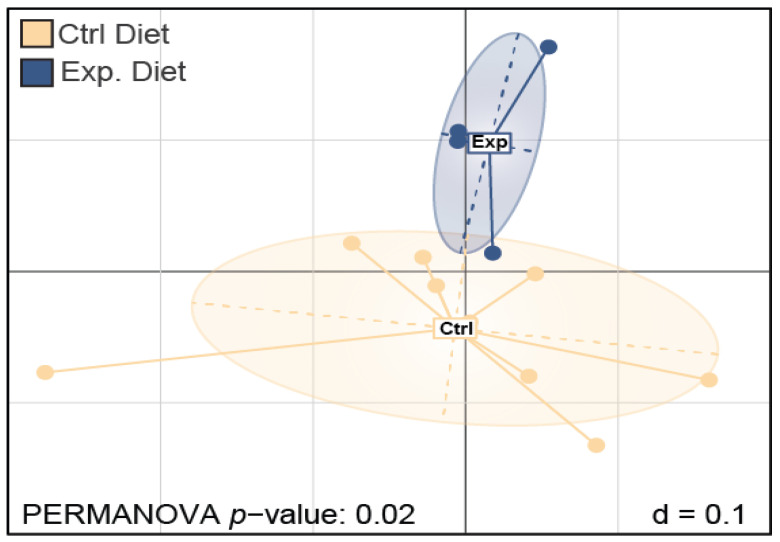
Multiple dimension scale (MDS) plots showing the diversity of the microbial profiles detected in kefalograviera cheese samples (control group, experimental group), considering the phylogenetic distances.

**Figure 4 foods-11-03164-f004:**
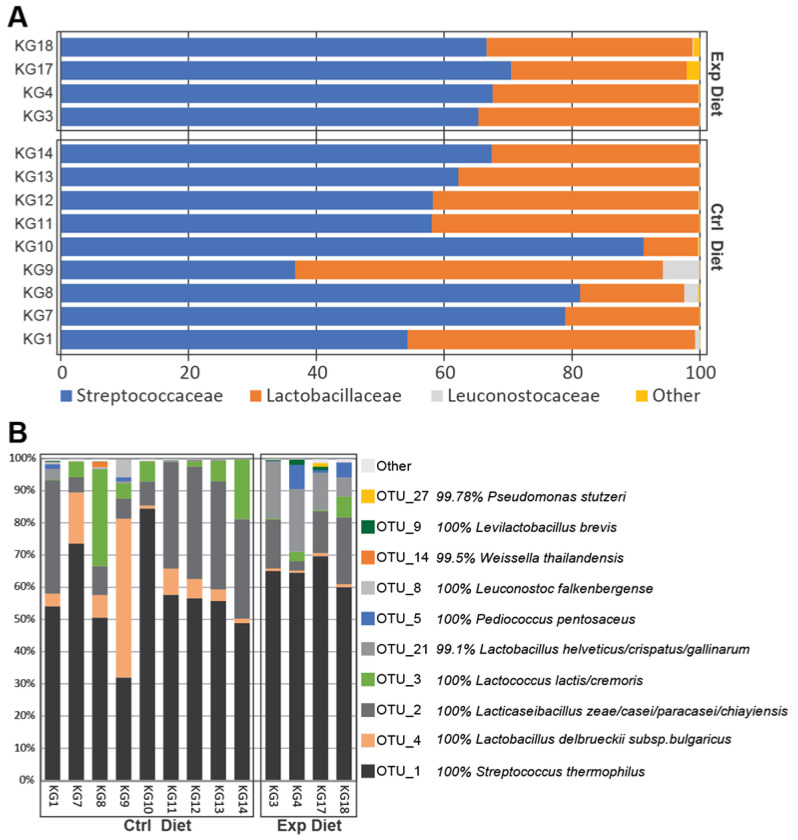
(**A**) Stacked barplots depicting the average microbiota composition of Kefalograviera cheese at family level for experimental and control cheeses samples. Data is presented as relative abundance on the overall microbial composition. Families below 0.5% relative abundance on average were grouped in the “Other” category, (**B**) Stacked barplots depicting the average microbiota composition of Kefalograviera cheese at species (OTUs) level for experimental and control cheese samples. Data is presented as relative abundance on the overall microbial composition. Only the main species-OTUs (Top 10) are presented. Low abundance species within each species are grouped in “Other” record. It is shown the percentage of similarity (%) identified by EzBioCloud’s 16S rRNA gene-based ID. *n* = 13; 9 control cheese samples and 4 experimental cheese samples.

**Figure 5 foods-11-03164-f005:**
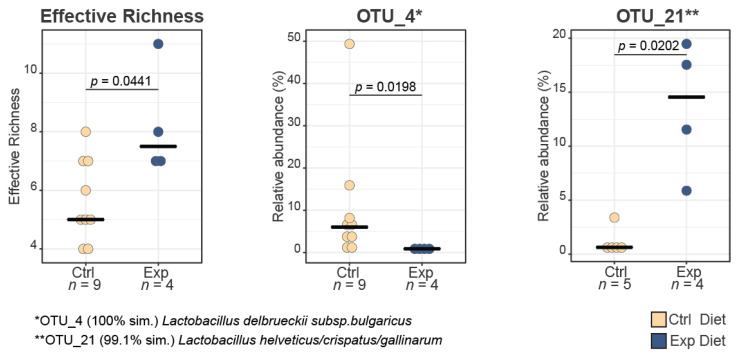
Boxplots of the effective taxonomic richness, OTU_4 and OTU_21 between control and experimental cheeses. Bold lines represent the median and whiskers account for the 95% confidence interval of the data. Outliers appear as circles. *n* = 13; 9 control cheese samples and 4 experimental cheese samples. *n* = 13; 9 control cheese samples and 4 experimental cheese samples.

**Figure 6 foods-11-03164-f006:**
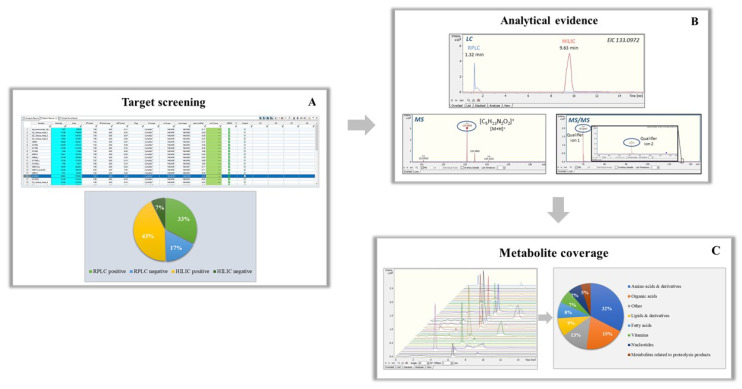
(**A**) Targeted approach for metabolites screening and percentage of detected compounds in each chromatography and ionization mode. (**B**) Example of L-ornithine detected in both RP and HILIC chromatography in positive ionization mode and its identification. (**C**) Metabolite profiling in Kefalograviera cheese samples; *n* = 24; 11 control and 13 experimental cheese samples.

**Figure 7 foods-11-03164-f007:**
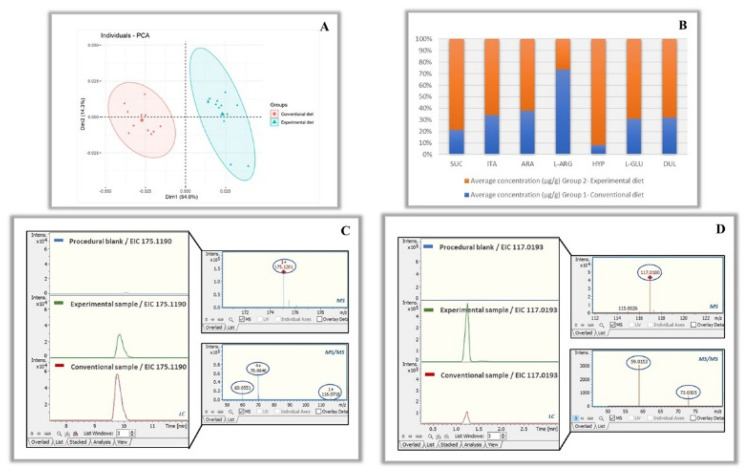
(**A**) PCA plot of the studied groups of Kefalograviera cheese. (**B**) Variation of significant metabolites between the two groups. (**C**) Example of L-arginine detected in higher concentrations in conventional samples. (**D**) Example of succinic acid detected in higher concentrations in experimental samples; *n* = 24; 11 control and 13 experimental cheese samples.

**Figure 8 foods-11-03164-f008:**
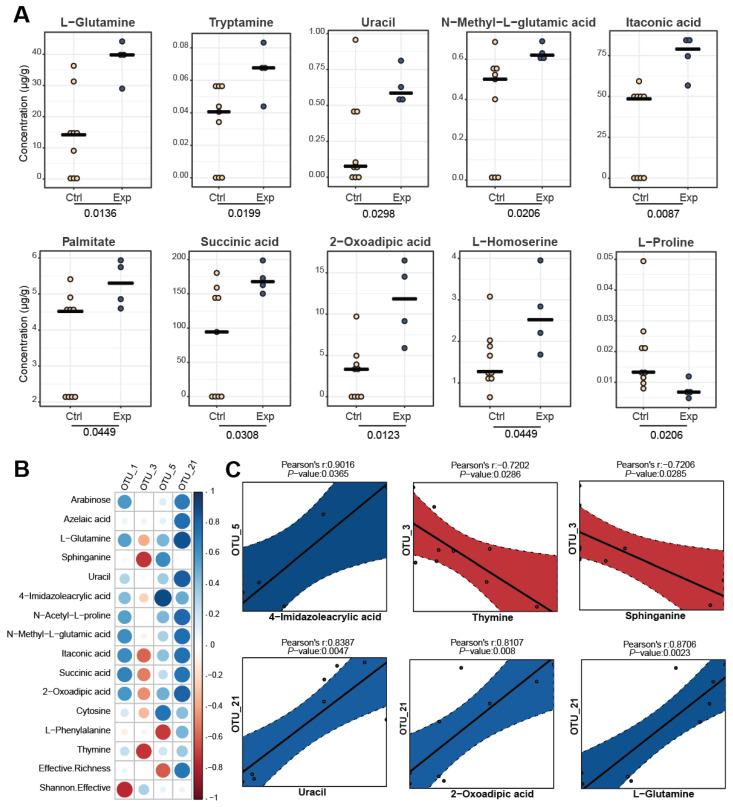
(**A**) Dot-plots of all significant comparisons (metabolites) between the two groups of Kefalograviera cheese samples (Ctrl: control group; Exp: experimental group). (**B**) Graphical display of the metabolites correlated with specific OTUs in a matrix. Each correlation is depicted as a small circle colored according to the direction of correlation coefficients (negative, red; positive, blue). The size of the circles is dictated by the uncorrected *p*-value of the corresponding correlation. (**C**) Correlations between each OTU and the metabolites. The graphs show the individual sample-specific values, a linear fitted line, and the lower and upper boundaries of the predicted interval (shown as a grey polygon around the fitted line). The boundaries are determined using the R function predict, which produces predicted values based on a linear model. A predicted interval accounts for the variability around the mean response inherent in any prediction. It represents the range where a single new observation is likely to fall. Due to the data transformation applied before calculation of correlations, there is no scale for the axes. The correlation coefficient and the original *p*-values are shown in each plot; *n* = 13; 9 control cheese samples and 4 experimental cheese samples.

## Data Availability

Primary sequencing data were uploaded to ENA-EBI public repository with the accession number PRJEB55879.

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
