# Peer review of "The “Crosstalk” between Microbiota and Metabolomic Profile of Kefalograviera Cheese after the Innovative Feeding Strategy of Dairy Sheep by Omega-3 Fatty Acids"

_foods, 2022, doi:10.3390/foods11203164_

Round 1

Reviewer 1 Report

Very interesting paper. The Authors used many modern methods and the results were presented in very interesting way. Results were analyzed very well statistically. The Authors  identified the key microbiota and metabolites characteristic for the Kefalograviera cheeses produced by sheep milk naturally enriched or not by omega-3 fatty acids. These results can be used to predict, optimize, and control cheese ripening and final quality of Kefalograviera cheese rich in omega-3 fatty acids. Obtained results can be used for cheese authenticity and traceability. The paper is well written and English language level is satisfactory.

I don't understand why the Authors do not mention their previous work on similar subject.

Tzora, A.; Nelli, A.; Voidarou, C.; Fotou, K.; Bonos, E.; Rozos, G.; Grigoriadou, K.; Papadopoulos, P.; Basdagianni, Z.; Giannenas, I.; et al. Impact of an Omega-3-Enriched Sheep Diet on the Microbiota and Chemical Composition of Kefalograviera Cheese. Foods 2022, 11, 843. https:// doi.org/10.3390/foods11060843.

This work should be compared and discussed with the results of reviewed paper.

Author Response

Point 1:

I don't understand why the Authors do not mention their previous work on similar subject.

Tzora, A.; Nelli, A.; Voidarou, C.; Fotou, K.; Bonos, E.; Rozos, G.; Grigoriadou, K.; Papadopoulos, P.; Basdagianni, Z.; Giannenas, I.; et al. Impact of an Omega-3-Enriched Sheep Diet on the Microbiota and Chemical Composition of Kefalograviera Cheese. Foods 2022, 11, 843. https:// doi.org/10.3390/foods11060843.

This work should be compared and discussed with the results of reviewed paper.

Response 1: Thank you for your comment. We have included the above reference in our introduction (Line 93-96) and discussion (Line 418-422) as recommended.

Reviewer 2 Report

This manuscript deals with differences of cheeses produced from milk of sheep fed different diets.

There are a couple of concerns about this study and its presentation.
There were only 8 animals that were given the experimental diet. It is difficult to draw wide-ranging conclusions from such a small population, which only yielded a total of 4 samples (apparently, but this number of four samples needs to be confirmed by the authors as it was not specified, but may be implied by the number of dots in Fig. 8A). Furthermore, the experimental diet is only minimally different from the control diet, in spite of their designation as 'soybean (control) or flaxseed and lupin diet (experimental)'. Both diets contain little soybean (~11% or 6%) or flaxseed (~3%) or lupin (~3%). There are also concerns about the LC/MS sample preparation, their usage of unpublished databases, among other issues (see below for details).

Some specific issues:
Lines 4, 31: Omega-3 fatty acids do not seem to have been analysed in this study, so reference to them in the title or abstract is misleading and should be deleted.
Line 16: Although the diets were designated as 'soybean (control) or flaxseed and lupin diet (experimental)', a closer examination of the diets in Table S1 reveals that both diets are based mainly on lucerne hay to ~45% and their soybean components are ~11% and ~6%, respectively, while the flaxseed and lupin diet only add ~6% to the total. Thus, to call them 'soybean (control) or flaxseed and lupin diet (experimental)' is misleading.
Line 44: Reference needed for statement.
Line 60: Define the abbreviation 'PDO'. All abbreviations need to be defined when first used.
Line 122-124: How many cheeses were made from sheep fed the experimental diet? Four?
Lines 225+231: As the authors cited reference 21, was EDTA and formic acid added just like in Fig.1 of reference 21, or was that step omitted? As for line 231, it it stated that the tubes were placed in 23 Degrees Celsius, but Fig.1 of reference 21, it is written 'minus 20 Degrees Celsius'. That is a big difference. Please clarify.
Line 276: Define the abbreviations PI and NI.
Lines 283-296: The usage of 'in-house metabolite databases makes it difficult for readers and reviewers to have confidence in the quality of the results. Are the authors willing to make their databases available to the scientific community?
Line 398. 'an increase of Pediococcus pentosaceus and Leuconostoc falkenbergense was additionally observed in the experimental cheese samples': This doesn't seem to be statistically relevant just by looking at Fig. 4B. Please present a more thorough analysis.
Line 407. 'increase in the effective richness': Please specify meaning.
Lines 465-519. Section 'Target metabolomic screening': Please show ion chromatographs of samples. Furthermore, it is unclear how the authors generated Figure S1. How did the authors quantify the concentrations in ug/g for the metabolites? It might be instructive to guide the reader through an example, such as succinic acid, that seems to have a variation between 'conventional diet' samples and 'experimental diet' samples.
Lines 470-472: the numbers of entries in their databases differ from what is mentioned in section 2.8.1. Please double-check the numbers.
Line 477. What kind of 'data analysis' do the authors refer to? Aren't the previous sentences already address some kind of data analysis?
Lines 533-534. What do 'comp 1', 'comp 2', 'comp 3', 'comp 4, 'comp 5, stand for?

Author Response

Thank you for your comments. 

Point 1:

There were only 8 animals that were given the experimental diet. It is difficult to draw wide-ranging conclusions from such a small population, which only yielded a total of 4 samples (apparently, but this number of four samples needs to be confirmed by the authors as it was not specified, but may be implied by the number of dots in Fig. 8A).

Response 1: We apologize for the confusion. Section 2.1 of the Materials and Methods has been edited to improve the presentation of the experimental design of this study. In particular, each treatment group consisted of 20 animals (n=40), which was the minimum representative number of dairy sheep of local breeds reared using the semi-intensive production system. This number of animals also ensured that bulk milk was gathered in adequate quantity for Kefalograviera cheese production. From the collected milk of each group, we produced 11 control and 13 experimental cheese samples (n=24). The cheese samples were obtained using industrial cheese-making practices and were from different production batches. This ensured the production of commercial-like cheese samples with a better representation of the microbiota and metabolome (compounds determining cheese taste and aroma) similar to the cheeses available to consumers. For the microbiota composition and correlation analyses, 9 control and 4 experimental cheese samples (n=13) were used to reduce the costs associated with the 16S rRNA sequencing. The higher quantity of control cheese samples was due to their use in another study. This information was also included in section 2.4 (Line 169-175) as well as in the description of Figures 2, 3, 4 A and B, 5 and 8A and B. For the metabolomic analysis, all cheese samples (n=24) were used as stated in Line 252-253 of section 2.6 and in Figure 6 A, B, C and Figure 7 A, B, C, D.

Point 2:

There are also concerns about the LC/MS sample preparation, their usage of unpublished databases, among other issues (see below for details).

Response 2: Thank you for your comments. Please check relative answers for Points 8 and 9.

Point 3:

Lines 4, 31: Omega-3 fatty acids do not seem to have been analysed in this study, so reference to them in the title or abstract is misleading and should be deleted.

Response 3: Thank you for your comment. We have updated Table S1 in the electronic supplementary material to include the fatty acid analysis of the control and experimental diet which shows the increase of omega-3 fatty acids in the experimental diet. Furthermore, we have edited the abstract (line 31) and the introduction (line 110-111,114) according to your suggestion.

Point 4:

Line 16: Although the diets were designated as 'soybean (control) or flaxseed and lupin diet (experimental)', a closer examination of the diets in Table S1 reveals that both diets are based mainly on lucerne hay to ~45% and their soybean components are ~11% and ~6%, respectively, while the flaxseed and lupin diet only add ~6% to the total. Thus, to call them 'soybean (control) or flaxseed and lupin diet (experimental)' is misleading.

Response 4: Thank you for this observation. We have edited the abstract (line 15-16) and the relevant part in section 2.1 (line 119-121) to describe our diets more accurately. In particular, the soybean-based diet was solely called control diet to minimize the confusion. Regarding your concern about the quantities of flaxseed and lupin in the experimental diet, the inclusion of these ingredients was the main difference between the two diets, as they were absent in the control diet.

Point 5:
Line 44: Reference needed for statement.

Response 5: Thank you for your comment. References have been added in this statement of the introduction (Line 43-44).

Point 6:
Line 60: Define the abbreviation 'PDO'. All abbreviations need to be defined when first used.

Response 6: Corrected as requested.

Point 7:
Line 122-124: How many cheeses were made from sheep fed the experimental diet? Four?

Response 7: This comment has already been answered in point 1.

Point 8:

Lines 225+231: As the authors cited reference 21, was EDTA and formic acid added just like in Fig.1 of reference 21, or was that step omitted? As for line 231, it it stated that the tubes were placed in 23 Degrees Celsius, but Fig.1 of reference 21, it is written 'minus 20 Degrees Celsius'. That is a big difference. Please clarify.

Response 8: Thank you for your kind notifications. As it is described in the manuscript, the sample preparation followed in this study, was an optimization of the preparation reported in Reference 21 in order to extract the metabolite content. The extraction step using EDTA and formic acid was omitted, as the addition of this chelating agent improves the extraction recovery of some antibiotics, especially of tetracyclines (as reported in the relative reference). This addition was not included as the targeted compounds were mainly endogenous metabolites and was not necessary for their extraction. The degrees were corrected (the tubed were placed at -23 oC, as in the reference). We apologize for the confusion.

Point 9:
Line 276: Define the abbreviations PI and NI.

Response 9: Thank you for your kind comment. The abbreviations are defined as:

PI: Positive Ionization

NI: Negative Ionization

The abbreviations were defined in Line 251-254.

Point 10:

Lines 283-296: The usage of 'in-house metabolite databases makes it difficult for readers and reviewers to have confidence in the quality of the results. Are the authors willing to make their databases available to the scientific community?

Response 10: Thank you for your kind comments. The in-house databases were built according to Mass Spectrometry Metabolite Library of Standards (MSMLS) (https://www.iroatech.com/mass-spectrometry-metabolite-library-of-standards-msmls/). The databases development workflow will be published online, individually, within next months. However, as it is very important to increase the confidence of the readers, database information regarding selected metabolites detected with significant difference between the two groups are given as an example, described in Supplementary Material (Table S3).

Point 11:
Line 398. 'an increase of Pediococcus pentosaceus and Leuconostoc falkenbergense was additionally observed in the experimental cheese samples': This doesn't seem to be statistically relevant just by looking at Fig. 4B. Please present a more thorough analysis.

Response 11: Thank you for your comment. We have edited this sentence to clarify that this observation is not statistically significant (Line 400-403). Regarding Figure 4B, it provides an overview of the average microbial composition between the control and experimental cheese samples with no statistical analysis being included.

Point 12:
Line 407. 'increase in the effective richness': Please specify meaning.

Response 12: We have included a reference in this line where the definition of this term is provided (Line 410-412). According to Reitmeier et. al 2021, “the count of taxa occurring above 0.25% relative abundance, referred to as “effective richness”.

Point 13:

Lines 465-519. Section 'Target metabolomic screening': Please show ion chromatographs of samples. Furthermore, it is unclear how the authors generated Figure S1. How did the authors quantify the concentrations in ug/g for the metabolites? It might be instructive to guide the reader through an example, such as succinic acid, that seems to have a variation between 'conventional diet' samples and 'experimental diet' samples.

Response 13: Thank you for your comments. Description of the quantification process was added to the Supplementary material (SM-2.1.), according to your recommendations. Relative reference was added to the manuscript in Line 295.

Point 14:
Lines 470-472: the numbers of entries in their databases differ from what is mentioned in section 2.8.1. Please double-check the numbers.

Response 14: We apologize for the mistake. Lines 501-504 were corrected according to the number of entries described in section 2.8.1.

Point 15:
Line 477. What kind of 'data analysis' do the authors refer to? Aren't the previous sentences already address some kind of data analysis?

Response 15: We apologize for the confusion. The Line 509 was corrected. Data analysis 4.3. Software was used for further process.

Point 16:
Lines 533-534. What do 'comp 1', 'comp 2', 'comp 3', 'comp 4, 'comp 5, stand for?

Response 16: As it is described in the manuscript, Principal component analysis (PCA), is a multivariate statistical procedure that allows to initially evaluate the information content in large data sets by means of a smaller set of “summary indices” that can be more easily visualized and analyzed. In our study, concentration data of the detected metabolites were used to evaluate the distribution of the samples. Principal components (PCs) are representing the extraction of important information from the data and expression of this information as a set of summary indices. The first four PCs, are being provided in Figure S2. The first principal component (PC1) represents the maximum variance direction in the data, while each observation may be projected in order to get a coordinate value along the PC-line. This value is known as a score. Along the same line, the PC2 reflects the second largest source of variation in the data while being orthogonal to the first PC. The two PCs can be visualized graphically, as represented in Figure 7 (a). As it is referred in Lines 568-572, taking into account the scores represented in Figure S2, the three PCs explain more than 40% variance. This is an important initial step to evaluate the potential success of any supervised methods, based on PC concept, such as PLS-DA models.

Round 2

Reviewer 2 Report

The authors have done an effort to clarify many concerns of this reviewer and significantly improved the presentation. This reviewers would still recommend showing ion chromatographs of the samples, as already mentioned in the review of a previous version of this manuscript.

Author Response

Point 1:

The authors have done an effort to clarify many concerns of this reviewer and significantly improved the presentation. This reviewer would still recommend showing ion chromatographs of the samples, as already mentioned in the review of a previous version of this manuscript.

Response 1:

Thank you for your kind feedback and recommendations. Ion chromatograms are being showed in Figure 7c, 7d for L-Arginine and Succinic acid, which are two of the metabolites significantly differentiated between studied groups. For these two compounds, a comparison between conventional (control) and experimental samples are being also represented. As reviewers recommended, additional extracted ion chromatograms in quality control (QC) samples were added in the Supplementary Material (Figure S1 in the revised version) for the selected important metabolites of the study (as also added their database information in the previous version).